# Soluble Heparin and Heparan Sulfate Glycosaminoglycans Interfere with Sonic Hedgehog Solubilization and Receptor Binding

**DOI:** 10.3390/molecules24081607

**Published:** 2019-04-23

**Authors:** Dominique Manikowski, Petra Jakobs, Hamodah Jboor, Kay Grobe

**Affiliations:** Institute of Physiological Chemistry and Pathobiochemistry and Cells-in-Motion Cluster of Excellence (EXC1003-CiM), University of Münster, D-48149 Münster, Germany; d.manikowski@uni-muenster.de (D.M.); petra.jakobs@googlemail.com (P.J.); jboor@hotmail.de (H.J.)

**Keywords:** heparan sulfate, hedgehog, heparin

## Abstract

Sonic hedgehog (Shh) signaling plays a tumor-promoting role in many epithelial cancers. Cancer cells produce soluble a Shh that signals to distant stromal cells that express the receptor Patched (Ptc). These receiving cells respond by producing other soluble factors that promote cancer cell growth, generating a positive feedback loop. To interfere with reinforced Shh signaling, we examined the potential of defined heparin and heparan sulfate (HS) polysaccharides to block Shh solubilization and Ptc receptor binding. We confirm in vitro and in vivo that proteolytic cleavage of the N-terminal Cardin–Weintraub (CW) amino acid motif is a prerequisite for Shh solubilization and function. Consistent with the established binding of soluble heparin or HS to the Shh CW target motif, both polysaccharides impaired proteolytic Shh processing and release from source cells. We also show that HS and heparin bind to, and block, another set of basic amino acids required for unimpaired Shh binding to Ptc receptors on receiving cells. Both modes of Shh activity downregulation depend more on HS size and overall charge than on specific HS sulfation modifications. We conclude that heparin oligosaccharide interference in the physiological roles of HS in Shh release and reception may be used to expand the field of investigation to pharmaceutical intervention of tumor-promoting Shh functions.

## 1. Introduction

In vertebrate embryogenesis, Sonic hedgehog (Shh)—one of the three members of the Hh family (Shh, Indian Hh, and Desert Hh)—acts as a morphogen and as a mitogen in various tissues at various developmental time points [1,2,3,4]. To fulfill these various roles, several unusual biochemical Hh properties have evolved that are only incompletely understood. The first unusual property is that all Hhs multimerize on the membrane of producing cells. This results in cell-surface clusters that consist of up to 50 19-kDa subunits. The second unusual property is that all Hhs are synthesized as dually lipid-modified molecules, which firmly tether to the cell membrane of the cells that produce them [5]. The first anchoring lipid—a cholesterol moiety—is covalently attached to the C-terminus during Hh secretion into the endoplasmic reticulum [6], and the second lipid—palmitic acid—is linked to an N-terminal conserved cysteine (C25 in mouse Shh) [7] by the activity of Hh acyltransferase (Hhat) [8,9,10,11]. The third unusual property is that, despite their multimerization and firm membrane association, Hh proteins initiate a response in cells at a significant distance from the source. This indicates that accessory molecules or mechanisms are required for Hh release and relay.

One such accessory mechanism is proteolytic Hh cleavage from both lipidated terminal peptides in a process called shedding [12,13,14,15]. Sheddases are soluble or membrane-bound proteases that solubilize the extracellular domains of various membrane-tethered proteins [16], thereby changing their localization and biofunction. Indeed, Hh shedding is one prerequisite for unimpaired Hh signaling during *Drosophila melanogaster* eye and wing development [15,17,18]. The N-terminal amino acid motif cleaved during Hh release, called the Cardin–Weintraub (CW) motif [19], also serves as a preferred binding site for heparan sulfate (HS) proteoglycans (HSPGs) [15,20,21,22]. This is important, as it suggests a possible key decision-making role of HSPGs in Hh release and bioactivation by binding to and blockading the CW sheddase target motif. In addition to this motif, HS/heparin can also interact with a basic residue located near the Hh binding site for its receptor [23,24]. This suggests a second possible decision-making role of HSPGs in the regulation of Hh reception on target cells.

HSPGs are ubiquitously expressed and consist of extracellular proteins to which linear HS chains are attached [25]. HS biosynthesis depends on the activity of several glycosyltransferases that add alternating N-acetylglucosamine (GlcNAc) and glucuronic acid (GlcA) residues in an unbranched fashion. The nascent chain undergoes specific modifications (sulfations and epimerizations) that are initiated by N-deacetylase/sulfotransferase family members. These bifunctional enzymes remove acetyl groups from GlcNAc residues, which are then sulfated by the N-sulfotransferase activity present on the same enzyme. The HS chain is further modified by a GlcA C5 epimerase, which converts GlcA into iduronic acid (IdoA) and 2-O, 3-O, and 6-O sulfotransferases. Together, these activities result in negatively charged HS chains that dynamically bind to patches of positively charged amino acids at the surface of several proteins [26,27,28], including the Hhs. Heparin constitutes the most highly sulfated form of HS, containing up to 2.4 sulfate groups per disaccharide, while most HS contains ~1 sulfate group per disaccharide [29]. The relative amount of IdoA in heparin is also increased over that in HS [30], while the extent of structural heterogeneity observed in HS is usually greater than that of heparin [31]. Finally, both heparin and HS show a broad molecular weight distribution, with an average molecular weight of ~30 kDa for HS and ~15 kDa for heparin.

Several aspects of cancer biology—including tumorigenesis, tumor progression, and metastasis—depend on HSPGs, which often regulate autocrine and paracrine signaling loops [32]. Clinical evidence indicates that pharmacological doses of heparin can have a marked effect on tumor growth and metastasis [33]. Moreover, when mutated or misregulated, Hh signaling can also contribute to tumorigenesis [34,35,36,37,38,39]: About 25% of cancer-related human deaths show signs of aberrant Hh signaling activation [40]. Such aberrant Hh signaling is associated with three types of oncogenic mechanisms: The Type I ligand-independent (autonomous) Hh pathway, the Type II ligand-dependent autocrine/juxtacrine Hh pathway, and the Type III ligand-dependent paracrine Hh pathway. Type I Hh signaling is activated independent of extracellular Hh through genetic alterations (mutations, amplifications, or deletions) in the Hh receptors Patched (Ptc) and Smoothened, or through downstream signal-transducing proteins, such as the glioma-associated oncogene (Gli) family of transcription factors [41]. One example of Type I cancer is basal cell carcinoma. Type II ligand-dependent activation of the cells of Hh origin, or of surrounding cells has been reported in malignancies such as pancreatic, esophageal, and stomach cancers, as well as in breast and colorectal cancers [38,42,43,44]. Type III cancers include cases of basal cell carcinoma, medulloblastoma, digestive tract tumors, and prostate cancer [38,45,46,47]. Shh signaling is also important for driving the self-renewal of cancer stem cells, a small subset of cells in a tumor that are able to initiate tumor spread and are resistant to chemotherapy [39,48]. These different malignancies call for the identification and targeted inhibition of mechanisms that drive extracellular Hh function [33,49]. On the basis of the known strong interaction between HS and Shh, we explored the potential of soluble heparin and HS derivatives to reduce Shh release from producing cells, as well as its binding to Ptc on responding target cells. We also aimed to define the structure/function relationship of HS/heparin interactions to expand the field of investigation to the pharmaceutical intervention of Hh-driven cancers.

## 2. Results

### 2.1. A “Hotspot” for Hh Binding to Ptc and Hh Activity Regulators

Structural analyses revealed the presence of a tetrahedrally coordinated Zn^2+^ cation in all vertebrate Hhs (Figure 1A, black sphere) [50,51]. This ion, together with amino acid residues in its proximity, interacts with the Ptc receptor (Figure 1A, red and cyan) [52] and with a Ptc-receptor antagonist called Hh-interacting protein (Hhip, Appendix A) [53,54]. The monoclonal anti-Shh antibody 5E1 [55] (Figure 1A, green) also binds to the Zn^2+^-coordinating groove, consistent with (and explaining) 5E1 blocking Shh interactions with Ptc and thereby diminishing Shh signaling (Appendix A). Notably, the Shh Zn^2+^ coordination site is also blocked by unprocessed N-terminal peptides in a protein called Shh^C25S^, again preventing its binding to Ptc (Figure 1B, Appendix A) [15]. Importantly, the HS-binding CW motif [15,20,21,22] is part of the unprocessed N-terminal peptide (Figure 1C), and proteolytic processing of this motif is required to remove the inhibitory peptide and to make Ptc binding possible. This couples Shh solubilization from the producing cell with its activation in vitro [15] and in vivo [17,18], and therefore suggests that HS-inhibited Shh shedding is one perceived strategy to reduce Shh biofunction (Figure 1E, bottom, scissor).

Of note, the same Zn^2+^-coordinating Shh “hotspot” for interprotein interactions, if exposed after Shh shedding, can also directly interact with HS [24] via residues K88, R124, R154, R156, and K179 (mouse Shh nomenclature, basic patch in Figure 1D,E, top left) [23,24]. This suggests that HS may control paracrine signaling by binding to, and blocking, the Ptc-receptor-binding site of activated Shh (Figure 1D,E). To test this possibility, we characterized the HS-modulated receptor binding of cancer-cell-produced Shh.

### 2.2. Proteolytic CW Processing Releases Shh In Vitro

Proteolytic processing stimulated by the soluble glycoprotein Scube2 (signal sequence, cubulin domain, epidermal growth factor-like protein 2) is one way to release Shh clusters from the surface of transfected Bosc23 cells, a HEK293 derivative [12,59,60]. To test whether the N-terminal HS-binding CW peptide K^33^RRHPKK^39^ is targeted in the process, we used bicistronic mRNA constructs for the coupled expression of Shh together with Hhat in the same cells. This strategy allowed for near-quantitative Shh palmitoylation and membrane association, resulting in near-quantitative N-terminal processing of the released protein [12]. Sodium dodecyl-sulfate polyacrylamide gel electrophoresis (SDS-PAGE) and immunoblotting was used to detect Shh in precipitated media and unprocessed precursor proteins in cell lysates (Figure 2A, Appendix A). As expected, Scube2 increased Shh solubilization about four-fold (Shh: 7% ± 5%, Shh + Scube2: 31% ± 12% (+440%), *p* < 0.01, *n* = 4). A Shh variant lacking all CW residues (Shh^Δ^) was not released, however, despite comparable cellular expression (Shh^Δ^: 2% ± 1%, Shh^Δ^ + Scube2: 2.5% ± 1%, *p* > 0.05, *n* = 4) (Figure 2A). This suggests that deletion of the CW site impairs Shh release from the cell surface. Notably, a Shh variant in which all five positively charged CW amino acids were replaced with neutral alanines (Shh^5xA^) was effectively released, even in the absence of Scube2 (Shh^5xA^: 88% ± 14%, Shh^5xA^ + Scube2 was always set to 100% and all other results expressed relative to this value, *p* > 0.05, *n* = 4). Therefore, a positive CW charge controls Shh processing and solubilization.

### 2.3. Proteolytic CW Cleavage is Essential for Hh Biofunction In Vivo

We confirmed the essential CW function in Hh solubilization by *Drosophila* driver lines *GMR45433*, *GMR45105*, and *GMR48462* in vivo. These flies overexpress recombinant Hh in tissues located inside and outside of the developing wing disc proper (Figure 2B, green) [61]. This resulted in wing overgrowth of anterior wing tissue resembling a natural *hh* gain-of-function allele, *hh^Moonrat^* [62] (Figure 2C, arrows). Using these developmental gain-of-function wing phenotypes as a readout for Hh biofunction, we found that the expression of Hh^3xA^ (in this variant, all three basic CW amino acids in fly Hh are replaced with neutral alanines) resulted in wings comparable to those formed as a consequence of ectopic Hh overexpression, demonstrating their activity. In contrast, we observed complete biological inactivity of Hh^Δ^ (this variant has the entire CW motif deleted) in accordance with its impaired solubilization in vitro (Figure 2A). Taken together, these data support the idea that the CW motif can be proteolytically targeted during Shh release and that impairing this process abolishes all Hh biofunction. The alternative explanation—that loss of Hh biofunction was due to the lack of CW-mediated HS binding—can be ruled out by the unimpaired biofunction of Hh^3xA^ that lacks all three HS-binding residues. Note that N-terminal Shh processing during release is not Bosc-cell specific, because HeLa (adenocarcinoma), Panc1 (pancreatic carcinoma), and MiaPaCa (pancreatic carcinoma) cells can also process the N-terminal Hh peptide during release from the producing cell surface, as indicated by its increased electrophoretic mobility (Appendix A).

### 2.4. Heparin Competes with Proteolytic Shh Processing and Release from Cancer Cells

As shown in Table 1, Panc1 cells express Shh and all necessary components for Shh release, most important, the essential Hh-releasing protein Dispatched (Disp) [63,64,65], A Disintegrin And Metalloprotease (ADAM) family members 10 and 17, and glypican HSPGs as potential release modulators [15,21]. We confirmed the biofunction of processed Panc1-derived Shh in C3H10T1/2 multipotent precursor cells that are widely used as a sensitive cell-based bioassay for Shh biofunction [66]. These cells express Ptc and Gli proteins required for Hh signal reception (Table 1) and, in the presence of Panc1-expressed Shh, differentiate into alkaline phosphatase (AP)-producing osteoblasts (Figure 3A). Specific C3H10T1/2 differentiation dependent on Shh activity was verified by the Shh-neutralizing monoclonal antibody 5E1 that blocks the Ptc-receptor-binding site [67] (Figure 1A) and the Shh pathway inhibitor cyclopamine [68] (Shh: 1.9 ± 0.01 arbitrary units (a.u.), *p* < 0.001, compared with Shh + 5E1: 0.3 ± 0.01 a.u., Shh + cyclopamine: 0.1 ± 0.002 a.u., and mock: 0.13 ± 0.02 a.u., *n* = 2).

To analyze whether heparin modulates N-terminal Shh processing, we cultured Panc1 cells in serum-free media for 6 h, harvested the media, ultracentrifuged it to remove cellular debris, and precipitated released Shh by using trichloroacetic acid. This was followed by SDS-PAGE and immunoblotting of cell lysates and precipitated media, as described earlier. As expected, in the absence of heparin, the cell-bound morphogen converted into truncated soluble Shh (Figure 3B bottom bands, asterisk, compare with Appendix A), consistent with Disp and ADAM expression in Panc1 cells (Table 1). Consistent with heparin binding to the Shh CW motif [22,71], soluble heparin impaired Shh processing and solubilization in a dose-dependent manner (Shh alone [no exogenous heparin] was set to 100%, *n* = 8; *p* < 0.05 compared with Shh + 1 μg/mL heparin: 77% ± 8%, *n* = 5; *p* < 0.01 compared with Shh + 2 μg/mL heparin: 54% ± 16%, *n* = 5; *p* < 0.01 compared with Shh + 5 μg/mL heparin: 26% ± 14%, *n* = 5; and *p* < 0.01 compared with Shh + 10 μg/mL heparin: 26% ± 15%, *n* = 5). This suggests that heparin blocks the HS-binding CW cleavage site. Alternatively, impaired Shh release may also be explained by heparin directly inhibiting Shh sheddase(s) at the cell surface. To distinguish between these possibilities, we expressed Shh^5xA^ that lacks all five HS-binding amino acids [71]. As expected, Shh^5xA^ release was only moderately affected by heparin (Shh^5xA^ was set to 100%, *n* = 10; *p* > 0.05 compared with Shh^5xA^ + 2 μg/mL heparin: 89% ± 14%, *n* = 5; with Shh^5xA^ + 5 μg/mL heparin: 93% ± 32%, *n* = 5; and with Shh^5xA^ + 10 μg/mL heparin: 88% ± 30%, *n* = 7) (Figure 3C). This result suggests that soluble heparin does not act on the Shh sheddase directly. Instead, it binds to, and directly blocks, the N-terminal CW cleavage site and thereby impairs Shh solubilization. Finally, we confirmed dose-dependent heparin inhibition of Shh solubilization from Bosc23 cells and HeLa cells (Bosc23: Shh alone was set to 100%, *n* = 2; *p* > 0.05 compared with Shh + 1 μg/mL heparin: 67% ± 10%, *n* = 2; *p* < 0.05 compared with Shh + 2 μg/mL heparin: 25% ± 27%, *n* = 2; *p* < 0.05 compared with Shh + 5 μg/mL heparin: 25% ± 29%, *n* = 2; and *p* < 0.01 compared with Shh + 10 μg/mL heparin: 15% ± 6%, *n* = 4. HeLa: Shh alone was set to 100%, *n* = 2; *p* > 0.05 compared with Shh + 1 μg/mL heparin: 62% ± 37%, with Shh + 2 μg/mL heparin: 28% ± 16%, with Shh + 5 μg/mL heparin: 29% ± 8%, and with Shh + 10 μg/mL heparin: 14% ± 9%, all *n* = 2) (Figure 3D,E). These results suggest that heparin impairs Shh release in a cell type-independent manner.

Unfractionated heparin has an average molecular weight of 15 kDa, and chain lengths can range from 8 to 25 kDa. To determine the inhibitory potential of smaller defined heparin oligosaccharides, we performed the same assays by using dp12 heparin (dp representing degree of polymerization; dp12 consists of 12 sugar residues) and dp30 heparin. As shown in Figure 3F and Appendix A, dp12 and dp30 reduced Shh solubilization from Panc1 and HeLa cells in a concentration-dependent manner. Nevertheless, unfractionated heparin is more potent in blocking Shh release from Panc 1 cells compared with dp12 or dp30. This indicates more efficient Shh release inhibition by larger-sized and highly charged forms of HS that bind to, and block, the N-terminal CW cleavage site.

### 2.5. Heparin Competes with Ptc for Shh Binding

As described earlier, both HS and heparin interact with basic residues located near the Ptc binding site of Shh [23,24] (Figure 1C,D), suggesting a second mode of Shh activity regulation on receiving cells. To test this possibility, we added increasing amounts of soluble heparin to conditioned media from Shh-expressing Panc1 cells and determined the remaining Shh potency to induce C3H10T1/2 precursor cell differentiation into AP-producing osteoblasts (Figure 4A). Consistent with overlapping Shh binding sites for Ptc and heparin, we found significantly impaired Shh-dependent C3H10T1/2 differentiation in the presence of 2–20 μg/mL heparin (*p* < 0.001 in all cases compared with cellular C3H10T1/2 differentiation in the absence of heparin (1.2 ± 0.2 a.u., *n* = 12): Shh + 2 μg/mL heparin: 0.6 ± 0.01 a.u., *n* = 6; Shh + 5 μg/mL heparin: 0.5 ± 0.04 a.u., *n* = 6; Shh + 10 μg/mL heparin: 0.47 ± 0.02 a.u., *n* = 6; and Shh + 20 μg/mL heparin: 0.44 ± 0.02 a.u., *n* = 6). To test whether heparin directly inhibits Shh binding to Ptc, we conducted the same assay in the presence or absence of heparin but replaced Shh with the purine derivative purmorphamine, a Hh signaling agonist acting downstream of Ptc [72]. As expected, we found that heparin did not decrease, but instead increased, purmorphamine-induced C3H10T1/2 differentiation (purmorphamine was set to 100%, *n* = 5: purmorphamine + 2 μg/mL heparin: 165 ± 40%, *p* < 0.01, *n* = 9; purmorphamine + 5 μg/mL heparin: 156 ± 25%, *p* < 0.05, *n* = 9; purmorphamine + 10 μg/mL heparin: 157 ± 33%, *p* > 0.05, *n* = 3; and purmorphamine + 20 μg/mL heparin: 184 ± 31%, *p* < 0.01, *n* = 3) (Figure 4B).

Heparin consists of a continuous arrangement of N-sulfated disaccharides (GlcA or IdoA linked to GlcNS, 1% of total heparin disaccharides), disulfated disaccharides (2-O-sulfated IdoA linked to GlcNS, or GlcA/IdoA linked to 6-O-sulfated GlcNS, 26%), and trisulfated disaccharides (2-O sulfated IdoA linked to 6-O-sulfated GlcNS, 73%, Figure 3, bottom). As a consequence, 8- to 25-kDa heparin chains are highly sulfated, bearing ~2.4 sulfates/disaccharide. In contrast, vertebrate HS consists of multiple unsulfated N-acetylated, highly N-sulfated, and mixed N-acetylated/N-sulfated domains. For this reason, compared with heparin, the average HS charge in vertebrate tissues is reduced [70]. To test whether the inhibitory potential of HS on Shh signaling is charge dependent, we compared heparin with commercially available HS fractions isolated from pork mucosa (Figure 4C). Because 5 μg/mL heparin and HS effectively blocked Shh-dependent C3H10T1/2 differentiation (Figure 4A, Appendix A), this concentration was used in this and all subsequent assays. The first isolate—HSI—carries 0.7 sulfates/disaccharide; the second isolate—HSII—carries 1.2 sulfates/disaccharide; and the third isolate—HSIII—carries 1.4 sulfates/disaccharide. We found that 5 μg/mL heparin strongly inhibited cellular C3H10T1/2 differentiation, as observed earlier (C3H10T1/2 differentiation in the absence of HS/heparin was set to 100%, *n* = 16: Shh + 5 μg/mL heparin: 27 ± 4%, *p* < 0.001, *n* = 16) (Figure 4C). HS inhibited Shh-induced C3H10T1/2 differentiation in a charge-dependent manner: While HSI showed a moderate yet significant effect (Shh + 5 μg/mL HSI: 70 ± 20%, *p* < 0.001, *n* = 8), more highly charged HSIII was more inhibitory (Shh + 5 μg/mL HSIII: 38 ± 7%, *p* < 0.001, *n* = 8), and moderately charged HSII showed an intermediate inhibitory potential (Shh + 5 μg/mL HSII: 58 ± 13%, *p* < 0.001, *n* = 8). Finally, we determined that Shh-induced C3H10T1/2 differentiation was inhibited to comparable levels by heparin and by dp30 heparin, but to a lesser degree by dp12 heparin (Shh + 5 μg/mL dp30: 26 ± 2%, *p* < 0.001, *n* = 4; Shh + 5 μg/mL dp12: 48 ± 11%, *p* < 0.001, *n* = 8; mock (no Shh): 13 ± 1%, *p* < 0.001, *n* = 8) (Figure 4C). This shows that heparin chains not much smaller than 30 sugar residues inhibit Shh signaling most strongly.

### 2.6. Oligosaccharide Structure Determination for Shh Binding

To further characterize whether specific heparin sulfation and chain length determine its interactions with Shh, we used a glycan microarray analysis [73] to screen a library of 53 structurally defined HS oligosaccharides that were coupled to the glass slide via their reducing ends (Appendix A). These oligosaccharides ranged in size from dp4 (4-mer) to dp12 (12-mer) and in charge from zero to ~3 sulfates/disaccharide (Figure 5). Unlabeled Shh was added to the immobilized oligosaccharides and protein binding was visualized by primary antibodies directed against Shh and fluorescent-labeled secondary antibodies. Lastly, detected fluorescence on the glass slide was quantified and presented as a histogram of fluorescence intensity.

Using this technique, we observed significant Shh binding to highly sulfated heparin 6-mers containing NS, 2S, 3S, and 6S (>0.8 sulfates/sugar residue), but not to low sulfated forms lacking more than one of these modifications (<0.8 sulfates/sugar residue) (Figure 5A). The same preferred binding to highly sulfated heparin oligomers was observed when heparin 7-mers (Figure 5B), 9-mers (Figure 5C), and 12-mers (Figure 5D) were analyzed, strongly supporting that Shh binding mostly depends on overall charge. However, we also observed that, if compared to heparin 6-mers #33 and #39 (1.2 and 0.8 sulfates/monosaccharide, respectively), Shh binding to 6-mer #15 was strongly reduced, despite its higher degree of sulfation (1.4 sulfates/monosaccharide) (Figure 5A). Likewise, Shh interacted more strongly with heparin 6-mer #39 than with with 6-mers #35 and #37, despite their same overall charge (0.8 sulfates/monosaccharide), and 7-mers #4 and #9 (1 sulfate/monosaccharide) also differed in their capacity to bind to Shh (Figure 5B). We explain preferred Shh binding to heparin 6-mers #33 and #39 by their dual-sulfated non-reducing ends. In contrast, non-reducing ends of heparin 6-mers #35, #37 and #15 were nonsulfated or monosulfated (Appendix A). This suggests that, due to their exposure away from the surface of the glass slide, non-reducing end sugars may critically contribute to (the initiation of) soluble Shh binding, especially in short oligosaccharides. Since non-reducing sugars in heparin-7-mers were all nonsulfated, some (structural) aspects independent of overall HS sulfation may also contribute to Shh binding.

We also observed that doubling the length of highly sulfated heparin oligosaccharides increased Shh binding by a factor of 5–10 (compare Figure 5A,D), depending on the type of sulfation. These results are in line with our previous analyses showing that Shh interactions depend both on heparin size (Figure 5E) and charge (Figure 5F). In contrast, we did not observe preferred Shh binding to any specific type of highly sulfated heparin: Oligosaccharides carrying NS, 2S, and 6S groups were always strongly bound, while N-sulfation in combination with only 2S and 6S was mostly insufficient (Figure 5C,F). Notably, when low-sulfated oligomers were compared, Shh binding to 2S/NS heparin oligomers was somewhat preferred over 6S/NS variants of the same length (compare oligosaccharides #42 with #44, and oligosaccharides #22 with #23, Figure 5F). This suggests that Shh binds highly sulfated HS rather unspecifically, resembling thrombin, which also binds HS solely on the basis of charge density. Therefore, Shh interactions with highly sulfated HS can be imagined as closely resembling a simple protein interaction with a cation exchange resin. Shh binding to shorter or low-sulfated heparin variants, however, may also prefer 2S groups over 6S groups and 6S/NS modification of the non-reducing sugar, at least under the experimental conditions used.

## 3. Discussion

The Hh pathway was discovered as a key pathway during embryonic patterning and development, but if misregulated, it can also contribute to various cancers in adults. Although Hh signaling in cancer has been extensively studied, most investigations have so far aimed at the characterization of small-molecule antagonists or transcription inhibitors in receiving cells to correct ligand-independent (Type I) Hh misregulation caused by mutations in the Shh receptor. Type II ligand-dependent Hh signaling can also be physiologically inhibited by the Ptc receptor antagonist Hhip [53] and non-physiologically inhibited by monoclonal antibody 5E1 [55] and the macrocyclic small molecule robotnikin [74,75]. All three molecules share the same mechanism of Shh signaling inhibition by binding to, and blocking, the Shh binding site for its receptor Ptc on the receiving cell surface [52,56]. The fifth known molecule class that binds to this site is HS [24], suggesting a possible mode of Shh activity regulation by a HS blockade of this site.

In this study, we experimentally confirmed this possibility and determined the HS size and charge requirements for the functional signaling inhibition of a Panc1-produced Shh variant on receiving cells. Consistent with Shh exhibiting a lower affinity for HS than for highly sulfated heparin [76], Shh biofunction was most strongly inhibited by unfractionated and dp30 heparin (ca. 6 kDa). Dp12 heparin (ca. 2.5 kDa) inhibited Shh-induced C3H10T1/2 differentiation to a lesser degree. HS also acted in a charge-dependent manner, with the most highly sulfated HSIII showing the strongest inhibitory effect and the less sulfated HSII and HSI showing more moderate effects. From these characteristics, we conclude that HS binding to Shh amino acids K88, R124, R154, R156, and K179 [23,24] modulates physiological Shh/Ptc interactions at the surface of receiving cells. In line with our results, several aspects of tumor cells are modulated by HS, including their onset and growth kinetics, invasiveness, and metastatic potential [77,78], and HS/Shh interactions are known to change the proliferation and invasiveness of Panc1 cells [23]. Thus, as shown for the monoclonal antibody 5E1 and robotnikin, Shh sequestration by soluble heparin may interrupt autocrine/paracrine Shh signaling loops, similar to what has been demonstrated for fibroblast growth factor 2 (FGF2) sequestration by HS to interrupt its interactions with the FGF receptor [79].

In this work, we also describe a second (indirect) regulatory role of HS as a competitor of proteolytic Shh processing and activation. Like Hhip, the monoclonal antibody 5E1, robotnikin, and HS, N-terminal unprocessed Shh peptides interact with, and block, the Ptc receptor binding “hotspot” prior to Shh release [12,15,60]. Proteolytic processing relieves this restriction in vitro and in vivo, and a prerequisite for N-terminal peptide processing is that sheddases can target the CW site. Consistent with this, the transfected cancer cell lines Panc1, HeLa, and MiaPaCa can release bioactive, N-truncated Shhs from their cell surfaces. Finally, sheddases and the shedding regulator Scube2 are expressed in malignant tumors and can participate in the pathology of these cancers [80,81,82].

Therefore, as a proof-of-concept, we confirmed that soluble heparin impaired N-terminal Shh processing and release by binding to, and blocking, the CW processing site. Unchanged levels of solubilized Shh^5xA^ impaired in its ability to bind heparin confirmed this idea and suggested that the CW-dependent initial step of Shh association with HSPGs on the membrane of producing cells [83] controls its release [59]. The reduced and variable inhibiting potential of HSI-III further suggests that efficient CW binding and blockade requires a high negative heparin charge. We confirmed this idea by using an array of chemically defined oligosaccharides spotted on a glass chip to assess binding of unprocessed Shh as a ligand: Here, as observed in our functional assays, increasing oligosaccharide length and charge correlated with increased Shh binding. Probes lacking 2-O sulfates or 6-O sulfates were less efficient binders, and non-sulfated glycosaminoglycans did not bind Shh. Taken together, these observations suggest that Shh/HS interactions depend on a high degree of N, 2-O, and 6-O sulfation, but less so on any specific type of sulfation. This is in contrast to other established HS-binding proteins, such as FGF2 or antithrombin. FGF2 requires contact with specific GlcNS and 2S, as well as an IdoA that causes a “kink” in the oligosaccharide binding sequence [84,85], and antithrombin binds specifically to a defined pentasaccharide sequence [86] only present in oligosaccharides #15, #17, and #18 in our assay. Our observed difference in the requirements for protein/HS interactions may reflect the need for strictly regulated HS-dependent protein activation/signaling, as in the cases of antithrombin and FGF2 versus HS roles in the assembly of multimeric Shh protein complexes [59,83] and in their release and transport [21]. We note that, because of their small size and extracellular function, heparin oligosaccharides can easily reach and modify cellular and soluble Shh and may therefore constitute promising candidate molecules for the generation of tailored anticancer therapeutics.

Taken together, in this report, we describe that two of the multifaceted roles of HS in Shh biology are the control of Shh processing, release, and activation and the competition with Ptc to bind processed Shh. These dual inhibitory HS roles in Hh biology may open exciting avenues for the generation of Shh-tailored polysaccharide-based cancer therapeutics that, in contrast to 5E1 or robotnikin, can also act on the non-released inactive precursor at the cell surface.

## 4. Materials and Methods

### 4.1. Heparin and HS

Heparin, heparin oligosaccharides, and fractionated HS were obtained from Iduron (Alderley Edge, UK). Glycan microarray analysis was conducted by Glycan Therapeutics (Raleigh, NC, USA) with purified recombinant murine Shh (GenScript, Piscataway, NJ, USA). Immobilized Shh was detected by using polyclonal α-Shh antibodies (goat IgG; R&D Systems, Minneapolis, MN, USA) and secondary Alexa Fluor 488 donkey α-goat IgG (Jackson ImmunoResearch; Cambridgeshire, UK). Fluorescence was acquired by using a GenePix 4300 A scanner (Molecular Devices; San Jose, CA, USA).

### 4.2. Cloning and Expression of Recombinant Proteins

Shh was generated from murine cDNA (NM_009170) by PCR. PCR products (Shh nucleotides 1–1314, corresponding to amino acids 1–438 of Shh; nucleotides 1–594, corresponding to amino acids 1–198 of non-cholesteroylated, monomeric ShhN) were ligated into pDrive (Qiagen, Hilden, Germany), sequenced, and relegated into pcDNA3.1 (Invitrogen, Carlsbad, CA, USA) for the expression of secreted, cholesteroylated 19-kDa Shh. To increase Shh N-palmitoylation, we obtained Hhat cDNA (NM_018194) from ImaGenes, Berlin, Germany, and cloned it into pIRES (ClonTech, Mountain View, CA, USA) for bicistronic Shh/Hhat and Shh^HA^/Hhat co-expression in the same transfected cells. Shh^C25S^/ShhN^C25S^ and variant cDNAs lacking the entire coding region for the HS-binding CW motif or having the coding region for all five HS-binding amino acids replaced by alanines were generated by site-directed mutagenesis (Stratagene, La Jolla, CA, USA) and sequenced. Because of higher expression and release rates, difficult-to-transfect cancer cell lines expressed C-terminally hemagglutinin (HA)-tagged Shh^HA^, unless otherwise indicated. *Drosophila melanogaster* Hh (NM_001038976) and variant cDNAs lacking the coding region for the HS-binding CW motif or with the coding regions for all three HS-binding amino acids replaced by coding regions for alanines were cloned into pUAST-attP (Invitrogen).

### 4.3. Cell Culture and Protein Analysis

Bosc23, MiaPaCa (ATTC CRM-CRL-1420), Panc1 (ATTC CRL-2553), HeLa (ATTC CRM-CRL-2), Capan1 (ATTC HTB-79), and mouse melanoma B16-F1 cells were cultured in Dulbecco’s modified Eagle’s medium (DMEM; PAA Laboratories, Cölbe, Germany) supplemented with 10% fetal calf serum (FCS) and 100 µg/mL penicillin–streptomycin. Bosc23 cells were obtained from D. Robbins, University of Miami, Florida, USA, and tested negative for mycoplasma. Cells were transfected by using PolyFect (Qiagen; Hilden, Germany) or ViaFect (Promega; Madison, WI, USA). Transfected cells were cultured for 36 h, the medium changed, and Shh secreted into serum-free media for 6 h. Where indicated, methyl-β-cyclodextrin (Sigma-Aldrich; St. Louis, MO, USA) was added at 100 μg/mL in serum-free DMEM. Proteins secreted into serum-free media were trichloroacetic acid-precipitated. All proteins were analyzed by 15% SDS-PAGE, followed by Western blotting with polyvinylidene fluoride membranes. Blotted proteins were detected by monoclonal α-HA antibodies (mouse IgG; Sigma, St. Louis, MO, USA), polyclonal α-FLAG antibodies (Sigma) directed against recombinant Scube2, polyclonal α-Shh antibodies (goat IgG; R&D Systems, Minneapolis, MN, USA), or polyclonal α-CW antibodies directed against the HS-binding CW sequence (rabbit IgG; Cell Signaling, Beverly, MA, USA). Incubation with peroxidase-conjugated donkey-α-goat/rabbit/mouse IgG (Dianova, Hamburg, Germany) was followed by chemiluminescent detection. Photoshop version CS5.1 (Adobe systems; San Jose, CA, USA) was used to convert grayscale blots into merged RGB pictures for improved visualization and quantification of N- and C-terminal peptide processing (α-Shh-detected proteins were always labeled green, α-CW-detected proteins red, and HA tags blue).

### 4.4. Shh Reporter Assays

C3H10T1/2 cells [66] were grown in DMEM supplemented with 10% FCS and 100 μg/mL penicillin–streptomycin. Twenty-four hours after seeding, Shh-conditioned media were mixed 1:1 with DMEM containing 10% FCS and antibiotics and applied to C3H10T1/2 cells in 15-mm plates. Purmorphamine (Sigma) served as a Shh agonist. To some samples, 2.5 μM cyclopamine—a specific inhibitor of Shh signaling—and 5 μg/mL of Shh neutralizing antibody 5E1 were added as controls to confirm the specificity of the assay. Generally, because of the variable expression levels, mutant and wild-type proteins were adjusted to comparable levels before induction of C3H10T1/2 differentiation. Cells were lysed 5–6 days after induction (phosphate-buffered saline, 0.5% TritonX-100, pH 7.4) and osteoblast-specific alkaline phosphatase (AP) activity was measured at 405 nm after the addition of 120 mM p-nitrophenylphosphate (Sigma) in 0.1 M glycine buffer, pH 9.5. Unless stated otherwise, assays were performed in triplicate.

### 4.5. Reverse Transcription Polymerase Chain Reaction (RT-PCR)

For RT-PCR analysis, TriZol reagent (Invitrogen; Carlsbad, CA, USA) was used for RNA extraction from various cultured cell types, and a first strand DNA synthesis kit (Thermo, Schwerte, Germany) was used for cDNA synthesis. PCR was performed by running 35 cycles using intron-spanning, species-specific primer pairs (sequences can be provided upon request). 

### 4.6. Fly Lines

The following fly lines were used: GMR-GAL4: GMR17G12 (GMR45433-GAL4): P(y[+t7.7]w[+mC]=GMR17G12-GAL4)attP2, Bloomington stock #45433 (discontinued but available from our lab); GMR18D10 (GMR45105-GAL4): P(y[+t7.7]w[+mC]=GMR18D10-GAL4)attP2, Bloomington stock #45105 (discontinued but available from our lab); GMR11E09 (GMR48462-GAL4): P(y[+t7.7]w[+mC]=GMR11E09-GAL4)attP2, Bloomington stock #48462 (discontinued but available from our lab). These lines were crossed with flies homozygous for UAS-hh or variants thereof. All Hh cDNAs cloned into pUAST-attP were first expressed in Drosophila S2 cells to confirm correct protein processing and secretion. Transgenic flies were generated by using the landing site 51C1 by BestGene. Cassette exchange was mediated by germ-line-specific phiC31 integrase [87].

### 4.7. Statistical Analysis

Pixel intensities on immunoblots were quantified with ImageJ (NIH, Bethesda, MD, USA). Relative protein release in a given experiment was calculated upon quantification of solubilized proteins divided by cellular expression and expressed as relative % (soluble protein/corresponding cellular protein). All statistical analysis was performed in Prism (GraphPad, San Diego, CA, USA) by using the Student’s t-test (two-tailed, unpaired, confidence interval 95%) or one-way analysis of variance tests (parametric, Bonferroni’s multiple comparison post-test, confidence interval 95%). All error estimates are standard deviations of the mean. 

### 4.8. Bioinformatics

The crystal structures of Shh (PDB: 3m1n, [56], PDB: 4c4n [24]) were displayed by using the PyMOL Molecular Graphics System, Version 1.3, Schrödinger, LLC (New York, NY, USA). 

## Figures and Tables

**Figure 1 molecules-24-01607-f001:**
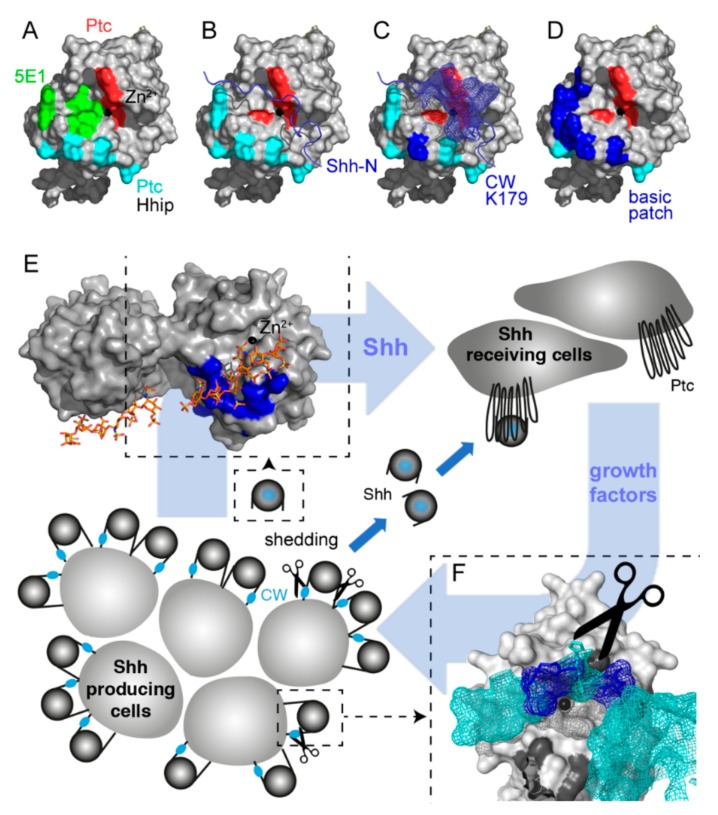
Multiple molecular interactions converge at the sonic hedgehog (Shh) pseudoactive site. (**A**) Shh contains a tetrahedrally coordinated Zn^2+^ cation (black sphere) that, together with amino acids H133, H134, H140, H180, and H182 (red), directly interacts with Hh-interacting protein (Hhip) and Patched (Ptc) peptides [52,53,54]. Cyan: amino acids K46, N51, R154, R156, and K179 that when mutated also affect Ptc binding [56]. The monoclonal antibody 5E1 binds to amino acids overlapping with the pseudoactive site (green) [55]. (**B**) In Hh clusters at the cell surface, unprocessed N-terminal peptides block the Shh binding site for Ptc (blue ribbon) [15,17]. (**C**) Unprocessed Shh clusters at the cell surface interact with heparan sulfate (HS) chains at two main sites: the basic N-terminal Cardin–Weintraub (CW) motif (mesh representation) and K179 (blue). (**D**) In addition to K179, processed bioactive Shh shows highly conserved basic amino acids in its vicinity (K88, R124, R154, R156, and K179, blue patch). (**E**) Model of two-way communication between tumor cells and their microenvironment (blue arrows). Tumors can secrete Shh that binds to Ptc receptors on stromal cells [35]. Stromal cells, including blood vessel cells, epithelial cells, fibroblasts, and immune cells, in turn support tumor growth by the secretion of other growth-promoting proteins [35,57]. Modified from [58]. Top left: clusters of basic amino acids at the molecular surface of soluble Shh can interact with HS (labeled blue are K88, R124, R154, R156, and K179; stick representation shows associated heparin in the structure). Analogous to Shh activity inhibition by MoAb 5E1, heparin/HS bound to this site may impair Shh binding to Ptc. (**F**) CW residues K32, R33, R34, K37, and K38 located in the N-terminal peptide of surface-associated Hh multimers (blue mesh) can also bind to HS. This would impair proteolytic processing of this site and Shh signaling to the tumor stroma. Shh: Sonic hedgehog, Ptc: Patched, Hhip: Hh-interacting protein, CW: Cardin–Weintraub motif.

**Figure 2 molecules-24-01607-f002:**
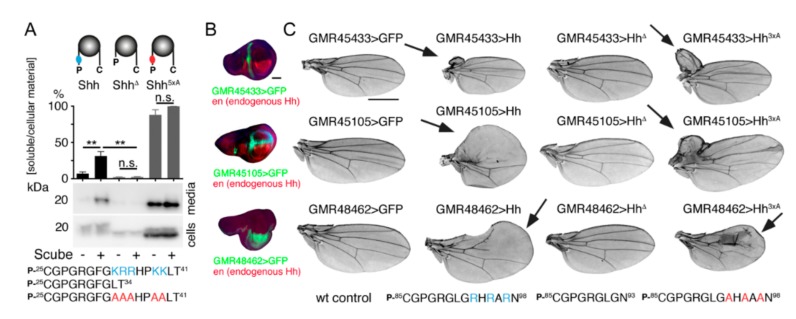
Proteolytic processing of N-terminal CW residues is a prerequisite for controlled Shh and *Drosophila* Hh release and biofunction. (**A**) Targeted deletion of the CW motif (Shh^Δ^) diminishes protein release from the cell surface, but a protein with all five basic CW amino acids (blue) replaced by alanines (red, Shh^5xA^) is strongly released even in the absence of Scube2. Shh^5xA^ + Scube2 was set to 100% and all other values are expressed relative to this value. One-way ANOVA; shown are average values ± SD, *n* = 4 for each data set. (**B**) UAS-CD8-GFP (green) produced in the anterior compartment or outside of the wing pouch under the control of *GMR45433* and *GMR45105*, or in the posterior compartment under *GMR48462* control. Cells in the posterior compartment produce endogenous Hh (red). Scale bar: 100 μm. (**C**) Adult *Drosophila* wings. In the normal wild-type situation (>CD8-GFP), the wing blade shows five longitudinal veins, an anterior cross vein, and a posterior cross vein. *GMR-Gal4*-induced Hh and Hh^3xA^ expression (note the presence of only three basic arginines in the fly CW-motif, labeled blue) causes anterior wing overgrowth or leads to overgrowth of the wing costa (arrows). In contrast, *GMR-Gal4* overexpression of Hh^Δ^ did not cause such defects, confirming biological inactivity of the expressed protein. Scale bar: 1 mm.

**Figure 3 molecules-24-01607-f003:**
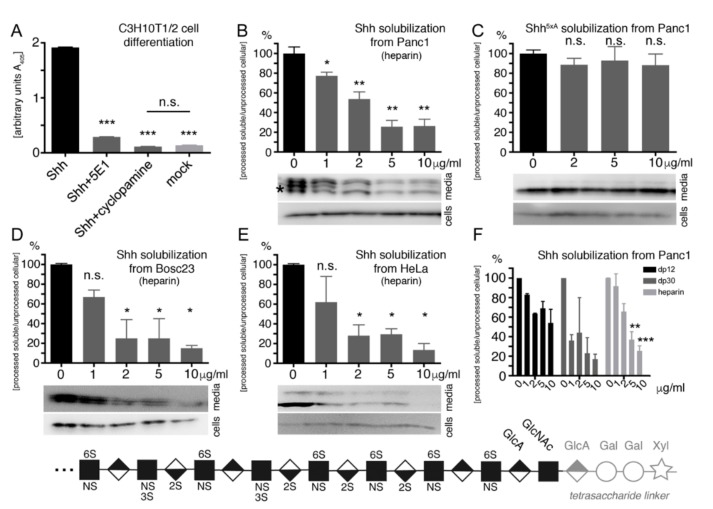
Heparin-modulated Shh release. (**A**) Shh expressed in Panc1 cells is bioactive, as indicated by Hh-dependent C3H10T1/2 reporter cell differentiation into alkaline phosphatase-producing osteoblasts. Shh biofunction was specifically inhibited by 5E1 and cyclopamine. *** denotes statistical significance (*p* < 0.0001, *n* = 2). (**B**) Increasing amounts of soluble heparin reduced Shh processing from Panc1 cells in a dose-dependent manner. * and ** denote statistical significance (*p* < 0.05 and *p* < 0.005, respectively, *n* = 5–8). (**C**) Impaired Shh release was specifically due to a blockade of the CW processing site because the release of Shh^5xA^ lacking all HS-binding basic amino acids remained unaffected. n.s.: not significant (*p* > 0.05, *n* = 5–10). Shh release from Bosc23 cells (**D**) and HeLa cells (**E**) was also inhibited by increasing heparin concentrations. (**F**) Small heparin oligosaccharides (dp12 and dp30) variably reduce Shh release from Panc1 cells. Bottom: heparin structure. Most acetyl groups from GlcNAc residues are replaced by sulfate groups to generate an extended N-sulfated (NS) domain. Extensive subsequent modifications, such as epimerization of GlcA to IdoA and 2-O, 6-O, and 3-O sulfations generate a highly sulfated, negatively charged region. Xyl: xylose, Gal: galactose, GlcA: glucuronic acid, GlcNAc: N-acetylglucosamine, IdoA: iduronic acid. Modified after [69,70].

**Figure 4 molecules-24-01607-f004:**
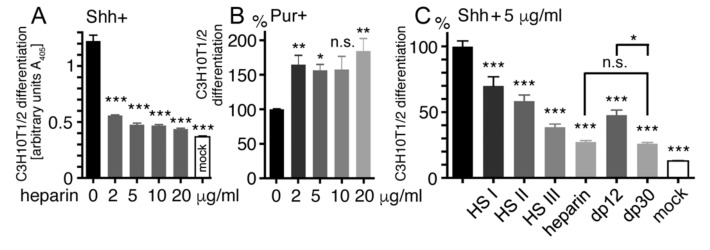
Soluble heparin impairs Ptc binding and biofunction of Shh. (**A**) Increasing amounts of soluble heparin were added to supernatants of Shh-expressing Panc1 cells. This significantly impaired Shh induced C3H10T1/2 precursor cell differentiation. *** *p* < 0.001 in all cases compared with C3H10T1/2 differentiation in the absence of heparin, *n* = 6–12. (**B**) Purmorphamine-dependent C3H10T1/2 differentiation is not impaired by 2–20 μg/mL exogenous heparin. ** *p* < 0.01, * *p* < 0.05, n.s. *p* > 0.05, *n* = 3–9. (**C**) Charge-dependent inhibition of C3H10T1/2 precursor cell differentiation. HSI carries the lowest charge density and heparin the highest charge density. Size-dependent inhibition of C3H10T1/2 precursor cell differentiation. Compared with that of heterogeneous heparin and dp30 heparin, the inhibitory potential of dp12 heparin is reduced. *** *p* < 0.001, * *p* < 0.05, *n* = 4–16.

**Figure 5 molecules-24-01607-f005:**
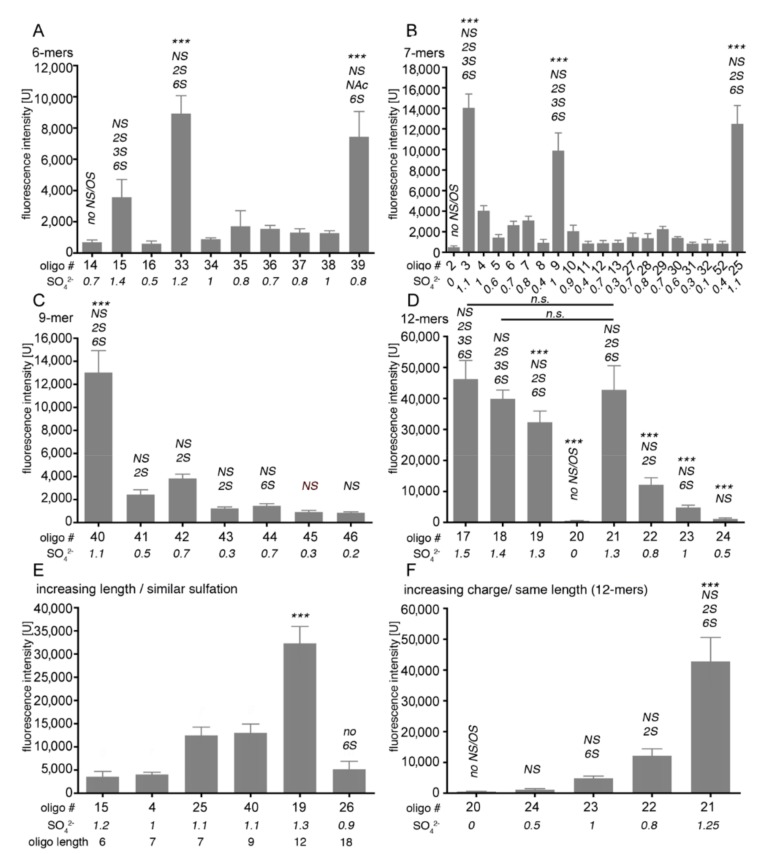
HS microarray analysis of Shh binding. A total of 53 HS oligosaccharides were printed on microarray chips (36 dots/oligosaccharide) and incubated with soluble Shh. Shh binding is expressed as a histogram of relative fluorescence intensity. The numbered oligosaccharide sequences and structures for each sample are listed in the Appendix A section; numbers in italics denote the average degree of sulfation per monosaccharide. (**A**) When 6-mers were compared, Shh binding to highly sulfated samples #33 and #39 was increased over that of all other samples (*p* < 0.001, *n* = 36). (**B**) Shh binding to 7-mers was also sulfation-dependent: Shh binding to samples #3, #9, and #25 was significantly increased over all other forms (*p* < 0.001, *n* = 36). (**C**) Sulfation-dependent Shh binding to oligosaccharides composed of 9 monosaccharides. Note that overall sulfation, but not any specific type of sulfation, is required for strong Shh binding (*p* < 0.001 when #40 is compared with all other samples, *n* = 36). (**D**) Sulfation-dependent Shh binding to oligosaccharides composed of 12 monosaccharides. All oligosaccharides differ significantly from each other in their Shh binding capacities (*p* < 0.001, *n* = 36), with the exception of samples #18 and #21 (n.s., *p* > 0.05) and #17 and #21 (*p* < 0.01, *n* = 36). (**E**) If oligosaccharides of different lengths but similar overall degrees of sulfation are compared, Shh binding to 12-mers is significantly increased over that of shorter forms. Longer oligosaccharides with reduced sulfation are less effective in Shh binding (compare #19 with #26). (**F**) If oligosaccharides of the same lengths but different overall degrees of sulfation are compared, Shh binding to highly sulfated probe #21 is preferred (*p* < 0.001, *n* = 36). Statistical significance was calculated by one-way ANOVA and Bonferroni’s multiple comparison test.

**Table 1 molecules-24-01607-t001:** Summarized semiquantitative detection of mRNA coding for proteins involved in Shh release from producing Panc1 cells or proteins involved in Shh signaling in receiving C3H10T1/2 cells. Required proteins for Shh assembly at the cell surface and (proteolytic) release are Dispatched (Disp), Glypican (Gpc) 1–6, and ADAM (A) sheddases 10, 12, and 17. Required proteins for Shh signaling on receiving cells are Patched (Ptc) and Gli family members 1–3.

	Disp1	Ptc	Shh	Gli1	Gli2	Gli3	Gpc1	Gp2	Gp3	Gpc4	Gpc5	Gpc6	A10	A12	A17
Panc1	+++	n.a.	+	-	-	+	+++	-	-	-	-	+	++	-	+++
C3H10T1/2	+	+	n.a.	-	+	++	+	-	-	++	-	++	n.a.	n.a.	n.a.

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
