# Peer review of "Soluble Heparin and Heparan Sulfate Glycosaminoglycans Interfere with Sonic Hedgehog Solubilization and Receptor Binding"

_molecules, 2019, doi:10.3390/molecules24081607_

Round 1
Reviewer 1 Report
In this paper the authors have defined that heparin (Hep)/heparan sulfate (HS) can block Sonic Hedgehog (Shh) solubilization and the receptor Patched (Ptc)-binding. And they also showed that the proteolytic cleavage of the N-terminal Cardin-Weintraub amino acid motif is a prerequisite for Shh solubilization and function. In addition, they demonstrated that HS/Hep binds to and block another set of basic amino acids, which were required for unimpaired Shh bindng to Ptc receptors. These results are novel and important for investigation of the tumor-promoting Shh functions. The following points are required to be improved before accepting for publication.
Maojor point #1.
To characterize HS/Hep for binding to Shh, several oligosaccharides with different structure were used for the interaction study. Based on the results, they concluded that Shh binds HS rather unspecifically. However, this is most likely an overhasty judgement. Based the results on Figure 5, the minimum size required for interaction seems to be hexasaccharide. The structural variety of a disaccharide unit is 2x2x2x2x2 (IdoA or GlcA, GlcNS or GlcNAc, +2S or -2S, +3S or -3S, and +6S or -6S) = 32. That of hexasaccharide unit is theoretically 32x32x32 = 32768. However, only 10 hexasaccharides were used in this study. When look at Figure 5A, it is obvious that the binding affinity is not correlated with the degree of sulfation. Oligo #15 has 1.4 sulfate/monosaccharide. Its binding to Shh is weaker than those of oligos #33 and 39, which have 1.2 and 0.8 sulfate/monosaccharide, respectively. Although oligos #35, 37, and 39 have the same degree of sulfation (0.8 sulfate/monosaccharide), only oligo #39 can bind to Shh. Thus, some binding preference seems to be observed. It is difficult to use a large number of structurally different oligosaccharides for interaction experiments and to obtain a conclusive nature for the binding from the present results. It might be the future perspective. However, some discussion is required.
Major point #2.
Table 1 is missing.
Author Response
The authors appreciate the positive comments made by the reviewers and constructive suggestions for improving the manuscript.
Reviewer 1 Comments for the Author...
1) To characterize HS/Hep for binding to Shh, several oligosaccharides with different structure were used for the interaction study. Based on the results, they concluded that Shh binds HS rather unspecifically. However, this is most likely an overhasty judgement. (…) When look at Figure 5A, it is obvious that the binding affinity is not correlated with the degree of sulfation.
We agree with the reviewer that, especially in the case of heparin 6-mers and 7-mers (Figure 5A,B), overall sulfation does not always correlate with their capacity to bind soluble Shh. In the case of heparin 12-mers, we also noticed preferred Shh binding to a low-sulfated variant carrying only NS and 2S, if compared to a low-sulfated 12-mer carrying NS only together with 6S (oligosaccharides #22 and #23, Figure 5F). In our revised manuscript, we now discuss these findings in detail (lines 350-362, lines 367-376). We also suggest preferred Shh binding to oligos #33 and #39 by their dual-sulfated non-reducing end sugars, because these sugars may be most accessible or may be involved in the initiation of Shh binding (because the reducing end sugars were used for oligo-coupling to the glass slide, as now stated in line 336, and may thus may be less accessible to the soluble protein than the free, outward-pointing end). Yet, we would like to note that the Shh binding profile is a lot broader than that of antithrombin that is used as a specific control protein in the assay. This protein binds only to 3S-carrying oligos #15, #17 and #18 (to a lesser degree to oligo#3). Therefore, we believe that our conclusion of relatively broad Shh binding still justified.
Yet, due to the observed preferences of Shh binding to oligos #33 and 39, we altered our conclusion of non-specific Shh/heparin interactions to interactions that “depend more on HS size and overall charge than on specific HS sulfation modifications” (lines 26-27) and “…and antithrombin binds specifically to a defined pentasaccharide sequence [86]only present in oligosaccharides #15, #17 and #18 in our assay”, now stated in lines 432-433.
2) Table 1 is missing.
We apologize for the mistake. The table is now added to the manuscript file.
Reviewer 2 Report
This manuscript entitled "Soluble Heparin and Heparan Sulfate Glycosaminoglycans interfere with Sonic Hedgehog Solubilization and Receptor Binding" examined the potential of defined heparin and heparan sulfate (HS) polysaccharides to block Shh solubilization and Ptc receptor binding. The results confirm in vitro and in vivo that proteolytic cleavage of the N-terminal Cardin–Weintraub (CW) amino acid motif is a prerequisite for Shh solubilization and function. Current manuscript was well written with self-explanatory data presenting. This reviewer recommends accepting this work after following minor revisions.
1. The resolution of Figure 1 should be improved.
2. Reference #1 and #24 need to be re-formatted if possible.
3. Reference #52 and #60 need more detailed info.
Author Response
The authors appreciate the positive comments made by the reviewers and constructive suggestions for improving the manuscript.
Reviewer 2 Comments for the Author...
1. The resolution of Figure 1 should be improved.
2. Reference #1 and #24 need to be re-formatted if possible.
3. Reference #52 and #60 need more detailed info.
We improved the resolution of figure 1 (the original file is now provided at a resolution of 300dpi). We also carefully checked the formatting of all references and updated references 52 and 60 (lines 699 and 720).
Reviewer 3 Report
In this manuscript the authors work to better understand how Shh solubilization, processing and signaling activation in recipient cells are finely regulated. The data support that soluble heparin and HS interfere with Shh solubilization and receptor binding by interacting with different motif and arise the possibility to investigate using heparin oligosaccharide to inhibit tumor-promoting functions. To have the conclusions to be more solid, mutation of 2nd HS binding site data may be needed too.
Major concern:
1. Fig.2A. It appears that deletion of the CW motif diminishes Shh expression, not related to Shh solubilization.
2. Fig.2. It is not clear why the in vitro study tested Shh5xA, and in vivo study tested Shh3xA ? Does Shh3xA function similar to shh5xA in vitro (Fig.2A) and in vivo ?
3. Fig.4 study. All these are heparin/HS inhibition analyses. Study with mutation in the 2nd HS binding site will greatly enhance the conclusion.
4. Fig. 5A and B appears to show some specific structure required (not sulfation level dependent: Oligos #15 vs # 33 vs#39, #4 vs.#9). Considering that the highly sulfated domain in HS is normally 4-6 mers. These 6-7 mers data might be more biologically relevant. The authors might at least to discuss this potential.
Minor concern:
1. The label “F” is missed in Fig. 1.
Author Response
The authors appreciate the positive comments made by the reviewers and constructive suggestions for improving the manuscript.
Reviewer 3 Comments for the Author...
1) Fig.2A. It appears that deletion of the CW motif diminishes Shh expression, not related to Shh solubilization.
As shown in Figure 2A, cellular expression levels of Shh and ShhDare similar. However, as correctly pointed out by the reviewer, Shh5xA expression is increased if compared to the other two forms. We do not know the reason for this, but would like to state that the relative amounts of solubilized ShhDis still much less if compared to the wild-type protein (that is expressed at comparable levels). Moreover, unlike the wild-type protein, ShhDrelease is not enhanced by the established Shh release factor Scube2. Therefore, we believe that our conclusion, that is, that ShhDrelease is most strongly affected by the mutation, is still correct.
2) Fig.2. It is not clear why the in vitro study tested Shh5xA, and in vivo study tested Shh3xA ? Does Shh3xA function similar to shh5xA in vitro (Fig.2A) and in vivo ?
Although the Cardin-Weintraub motif is present in all Hhs, the fly ortholog has only three basic residues instead of the five found in vertebrates. This is now stated in the revised manuscript (lines 177-178).
3) Fig.4 study. All these are heparin/HS inhibition analyses. Study with mutation in the 2ndHS binding site will greatly enhance the conclusion.
This reviewer raises an important point. Indeed, for a related project, we have already conducted scanning mutagenesis to delete potential HS-binding basic amino acids K46, K88, R102, R104, R124, R154, R156 and K179 alone or in combination (residues that in part were also suggested by others [1]). These proteins were then analyzed by HS- and heparin affinity chromatography, revealing that only the combined deletion of HS-binding amino acids significantly impairs HS- and heparin binding.
Although we agree with this reviewer that testing such mutant control proteins in our functional C3H10T1/2 assays, hoping that their differentiating ability would be independent of the presence or absence of exogenous HS, would provide additional evidence for specific Shh signaling inhibition by exogenous HS. However, such a functional experiment cannot be performed based on the shared role of residues K46, K88, R124 and R154 in Ptc-receptor binding [2]. Mutant proteins lacking these residues are impaired in their capacity to induce C3H10T1/2 reporter cells, yet retain their capacity to bind the most highly charged forms of HS used in our study. For this reason, we refrained from their analysis in the present study.
We would like to refrain from presenting any of the scanning mutagenesis data for the final reason that results of a follow-up study are currently under review at another journal. The scientific question asked in this study and the presented results are not related to the study presented here: In the other study, we deleted one particular residue (K179 [3]) not involved in Ptc-binding and studied the consequence of this mutation in vitro (HS-binding is only impaired to a minimal level) and functionally in Drosophila melanogaster. In vivo, we observed changes in the ability of the Hh mutant protein in HS-dependent long-range transport during wing development. This study can be provided upon request. We would therefore prefer to keep our mutagenesis study separate from the results presented in this manuscript.
1. Whalen, D. M.; Malinauskas, T.; Gilbert, R. J.; Siebold, C., Structural insights into proteoglycan-shaped Hedgehog signaling. Proc Natl Acad Sci U S A 2013,110, (41), 16420-5.
2. Gong, X.; Qian, H.; Cao, P.; Zhao, X.; Zhou, Q.; Lei, J.; Yan, N., Structural basis for the recognition of Sonic Hedgehog by human Patched1. Science 2018,361, (6402).
3. Chang, S. C.; Mulloy, B.; Magee, A. I.; Couchman, J. R., Two distinct sites in sonic hedgehog combine for heparan sulfate interactions and cell signaling functions. J Biol Chem 2011,286, (52), 44391-402.
4) Fig. 5A and B appears to show some specific structure required (not sulfation level dependent: Oligos #15 vs # 33 vs#39, #4 vs.#9). Considering that the highly sulfated domain in HS is normally 4-6 mers. These 6-7 mers data might be more biologically relevant. The authors might at least to discuss this potential.
We agree with the reviewer that some heparin 6-mers and 7-mers bind the soluble Shh ligand better despite their lower overall sulfation. This indicates some binding preference not related to overall sulfation (=charge), which is now noted and discussed in the revised manuscript (lines 350-362, lines 367-376). We also discuss one example of a specific HS-binder – antithrombin – that only binds to oligos #15, 17 and 18 in our assay (lines 432-433).
5) The label “F” is missed in Fig. 1.
We apologize for this mistake. The figure is now properly labeled.
Round 2
Reviewer 1 Report
This paper can be accepted for publication on this journal, since the manuscript has been well revised compare to the original version.